# Oocyte Exposure to Low Levels of Triclosan Has a Significant Impact on Subsequent Embryo Physiology

**DOI:** 10.3390/ijerph22071031

**Published:** 2025-06-28

**Authors:** Vasiliki Papachristofi, Paul J. McKeegan, Henry J. Leese, Jeanette M. Rotchell, Roger G. Sturmey

**Affiliations:** 1Centre for Biomedicine, Hull York Medical School, University of Hull, Hull HU6 7RX, UK; hyvp2@hyms.ac.uk (V.P.); henry.leese@hyms.ac.uk (H.J.L.); 2Hull York Medical School, University of Hull, Hull HU6 7RX, UK; paul.mckeegan@hyms.ac.uk; 3College of Health and Science, University of Lincoln, Lincoln LN6 7DQ, UK; jrotchell@lincoln.ac.uk

**Keywords:** triclosan, endocrine disrupting chemicals, embryo metabolism, oocyte metabolism

## Abstract

Triclosan (TCS) is an antimicrobial agent in a wide range of health care products. It has been found in various human bodily fluids and is a potential reproductive toxicant. However, the effect of TCS on early embryo development in mammalian species is limited. We therefore asked whether exposure to TCS affects mammalian cumulus–oocyte complexes (COCs), and if so, whether the effects persist into the early embryo. COCs, isolated from abattoir-derived bovine ovaries, were exposed to two environmentally relevant doses of TCS (1 and 10 nM) during in vitro maturation. When exposed to 1 nM TCS during in vitro maturation, progesterone release from bovine oocytes was elevated. Furthermore, altered pyruvate metabolism and mitochondrial dysfunction were also observed; specifically, O_2_ consumption coupled to ATP production was significantly decreased in COCs after acute exposure to TCS prior to maturation, whereas proton leak from the respiratory chain was increased. Subsequently, TCS-exposed COCs were fertilised. Fewer oocytes were able to develop to blastocyst when exposed to 1 nM TCS during maturation compared to the Control group, and those that did reach the blastocyst displayed impaired glycolytic and amino acid metabolic activity. These findings indicate for the first time that oocytes exposed to TCS during the final stages of maturation give rise to embryos with impaired mitochondrial function, altered steroidogenesis, and disrupted metabolic activity.

## 1. Introduction

Triclosan (TCS) is a broad-spectrum agent with antimicrobial activity, commonly found in products including soaps, healthcare antiseptic products, such as mouthwash, toothpaste, personal hygiene products, including acne cream, skin cream, sunscreens, and deodorant, and medical products, including surgical sutures, catheters, and ureteral stents. Although the use of TCS was restricted in certain wash products in the USA in 2016 (FDA, 2016) and from products used in hospitals in 2017 (FDA, 2017), the inclusion of TCS remains permitted in personal care products, cosmetics, clothes, and toys. Because of this, the daily TCS exposure of the general population is still estimated to be high, with a geometric mean from 0.7 to 7.85 μg/L depending on the country [1]. As a consequence of extensive inclusion in consumer products, TCS has been widely reported to have accumulated in the aquatic and terrestrial environment [2,3,4]. Such bioaccumulation in aquatic biota has prompted concerns about the impact of TCS on human and animal health [5].

Of particular concern, TCS has been reported to be present in a range of biological fluids. For example, TCS was detected at nanomolar concentrations in over 2000 urine samples collected during a study of the USA general population [6]. Similar reports have confirmed the presence of TCS and its conjugated forms in the urine of population cohorts from India [7], Spain [8], and Belgium [9]. Furthermore, in a study of 1890 first-trimester urine samples of pregnant women, free TCS and TCS-metabolites (with geometric means of 0.5 and 12.30 μg/L, respectively) were detected in more than 75% of samples examined [10]. Perhaps most startlingly, TCS has been identified in blood (4.1–41.4 nM; [8]), serum (0.035–1200 nM; [11]), and breast milk over a wide concentration range (0.062–252 nM) [8,12,13,14,15]). Additionally, TCS has been detected in human tissues, including adipose tissue, brain, and liver, at levels of 0.23–29.03 ng/g [15], and a recent study reported the presence of TCS in the ovarian follicle at a concentration of ~100 pg/mL [16]. Of critical importance, there are reports of correlations between increased urinary concentration of TCS (8.9–19.1 μg/L) and diminished ovarian reserve [17,18] and of follicular TCS levels and embryo quality [16], hinting at a link between TCS and fertility. Moreover, TCS has been reported to be an Endocrine Disruptor Compound (EDC; [19]), with clear effects on pregnancy in a number of animal models, as reviewed in [19].

Beyond studies of TCS accumulation uptake, the biological implications of such exposure are now becoming apparent. For instance, TCS exposure has been correlated with oxidative stress and mitochondrial dysfunction in porcine oocytes [20,21] and zebrafish embryos [22,23]. Indeed, a study in human epithelial cell lines showed that TCS (1.25–50 μM) could impair mitochondrial function by stimulating superoxidase release and inhibiting complex II [24], suggesting a direct effect of TCS on cellular metabolic processes. Furthermore, TCS exposure in the micromolar (μΜ) range seems to be correlated with impaired function of early reproductive events, such as reduced meiotic maturation in pig oocytes [21], and in the implantation process through an effect on uterine receptivity [25] and on preimplantation embryo development by reducing the expression of pluripotent markers [26] in mice.

Oocyte quality is a major determinant of embryo developmental competence. Disruption of COC physiological function can have subsequent effects on the physiology and metabolic function of subsequent early embryos, as reviewed by Krisher [27] and Keefe et al. [28]. This is important because impaired metabolic function in embryos has been correlated with decreased developmental outcomes in mouse [29,30], bovine [31], and human embryos [32,33]. Nutritional markers, such as glucose and pyruvate depletion and lactate production, and amino acid turnover [34,35], as well as physiological markers such as oxygen consumption rate, [36,37] have previously been associated with embryo quality and developmental potential. Underscoring this, recent evidence ties appropriate metabolic function to genome activation [38], illustrating the fundamental importance of metabolism to early development. Beyond this, the metabolic activity of embryos may have links to the lifelong health of the resulting offspring. The Developmental Origins of Health and Disease (DOHaD) hypothesis [39] has proposed that certain environmental stimuli (such as nutrition, stress, and chemical exposure) during essential developmental periods may induce phenotypic adaptation during the preimplantation stages of development that persist into later life [40,41,42].

Given the intrinsic importance of gamete and embryo metabolism, the aim of this study was to examine the extent to which steroidogenesis and metabolic activity of bovine COCs during in vitro maturation (IVM) are sensitive to the reproductive toxicant TCS. In addition, the embryo developmental potential and the metabolic profile of the resulting embryos were assessed at the blastocyst stage. The bovine model was chosen since it has been identified as an effective model for reproductive toxicological studies due to its functional and physiological similarities with humans [43,44]. Importantly, this study has focused on physiologically relevant doses of TCS.

## 2. Materials and Methods

All work was carried out after ethical review from the Institutional Ethical Review Board (Ref HYMS 18 47).

### 2.1. Experimental Design Overview

COCs, isolated from abattoir-derived ovaries, were cultured in the presence of two different doses of TCS (10 and 1 nM) during in vitro maturation. Both doses were within the range previously detected in human bodily fluids. The levels of steroid hormones (17β-estradiol (E2) and progesterone (P4)) and key metabolites (pyruvate, lactate) were quantified in spent media using ELISA and microfluorometric assays, respectively. Furthermore, using extracellular flux analysis, the oxygen consumption rate (OCR) was measured as an indicator of the mitochondrial function of COCs [45] acutely exposed to TCS prior to maturation.

Subsequently, the TCS-exposed COCs were fertilised and cultured for in vitro embryo production without further TCS addition. The rates of embryo cleavage and blastocyst formation were recorded on Day 2 and Day 7/Day 8, respectively, and the metabolic activity of the resulting blastocysts was determined. The key energy metabolites glucose, pyruvate, lactate, and amino acid turnover were quantified in spent media using microfluorometric assay and HPLC, respectively.

### 2.2. Cumulus–Oocyte Complexes’ In Vitro Maturation (IVM) and In Vitro Culture of Bovine Embryos (IVC)

Bovine reproductive tracts were collected from a local abattoir and transferred to the laboratory within 2 h of the animals’ slaughter. Upon arrival at the laboratory, the ovaries were isolated from the rest of the reproductive tract and washed three times in pre-warmed PBS supplemented with antimycotic-antibiotic at 39 °C. IVM and IVC were performed as described previously [46]. Briefly, follicles with a diameter of 2–8 mm were punctured and the contents aspirated into M199 media supplemented with HEPES buffer, heparin, and bovine serum albumin (BSA). Oocytes with at least 2 layers of intact cumulus cells were selected, washed, and placed in groups of 35–45 into 500 μL bovine maturation media, which was prepared in-house from M199 media supplemented with NaHCO₃, gonadotrophins FSH and LH, and growth factors EGF and FGF. COCs were incubated in a humidified environment, in 5% CO_2_, in air for 21 to 23 h. After this time, mature COCs were used for in vitro fertilisation (IVF) and the spent maturation medium was stored at −80 °C until further analysis, as described in the following sections.

For IVF, semen samples stored in liquid nitrogen, from a single bull with proven fertility, were thawed instantly in a warm bath at 39 °C. The semen samples were transferred in Semen-Prep commercial media (IVF Bioscience) and centrifuged twice at 328× *g* for 5 min. Groups of 35–45 oocytes were co-incubated with 1 × 10^6^ sperm cells in fertilisation-TALP media for 18–22 h in 5% CO_2_, in air.

Putative zygotes were selected and vortexed vigorously for 2 min to remove any remaining cumulus cells before being placed in groups of 20 to 25 in 30 μL of Synthetic Oviduct Fluid media supplemented with amino acids and BSA (SOFaaBSA) under mineral oil and incubated for maximum of 8 days under hypoxic conditions (5% CO_2_, 5% O_2_ Bal N_2_). On Day 2 and Day 7/day 8, cleavage and blastocyst rates of the embryos were recorded, respectively. On Day 7 and Day 8, the blastocysts were retrieved for individual embryo culture described in the following section.

### 2.3. Preparation and Storage of TCS Stocks

For the purposes of all experiments—except for those of oxygen consumption rate measurement—TCS (CAS: 3380-34-5) was diluted into 100% DMSO (CAS: 67-68-5) creating a stock of 500× the final concentration and stored in aliquots at −20 °C in glass vials for a maximum of 3 months in order to prevent TCS degradation. On the day of IVM, the TCS stock was further diluted into maturation medium for a final concentration of 10 and 1 nM and incubated in 5% CO_2_, in air at 39 °C for at least 2 h prior to COCs addition. TCS exposure took place solely during oocyte maturation. Where appropriate, vehicle controls were prepared with DMSO added to the maturation medium in the absence of TCS.

### 2.4. Individual In Vitro Culture of Embryos for Metabolic Assays

On Day 7 and Day 8 of embryo culture, blastocysts were washed twice and then cultured individually in SOF “analysis” media in 5 μL droplets under oil for approximately 22–24 h in hypoxic conditions of 5% CO_2_ and 5% O_2_. SOF “analysis” medium is an identical version of SOFaaBSA in all aspects except for glucose and lactate concentrations, which are 0.5 and 0 mM, respectively [31]. The morphological stage and precise time of transfer of blastocysts into and out of the SOF analysis medium were recorded. At the conclusion of embryo culture, the blastocysts were removed, and the incubation plates were sealed and stored at −80 °C until analysis. Finally, consumption and release analysis (CORE analysis) of key metabolites and amino acids was performed as described in [31].

### 2.5. Determination of Steroid Hormone Release

Spent media from COCs exposed to TCS during maturation were collected and stored at −80 °C for E2 and P4 quantification. E2 and P4 concentrations in the spent media were quantified using enzyme immunoassay reagents (Enzo Life Science, Farmingdale, NY, USA). The reported sensitivity of the E2 ELISA kit (ADI-900-008) was 28.5 pg/mL, and the inter-assay and intra-assay coefficients of variation ranged from 5.2 to 7.4% and 8.4 to 9.2%, respectively. The limit of detection of the P4 ELISA kit (ADI-900-01) was 8.57 pg/mL, and the inter-assay and intra-assay coefficients of variation varied from 2.7 to 6.8% and 4.9 to 7.6%, respectively. The quantification of hormone concentrations in the samples was performed according to the manufacturer’s instructions and subsequently normalised against the number of COCs and the hours of incubation per spent media sample, expressed as pmol/hour/COC.

### 2.6. Identification of COCs Nuclear Status After IVM

COCs were collected after IVM, washed three times in 0.2% PVP-PBS, and then vortexed vigorously to remove remaining cumulus cells. Finally, the denuded oocytes were labelled with Hoechst-EtOH nuclear staining and incubated at 4 °C overnight, after which the nuclear status was assessed using a fluorescence microscope (Axiocam 506, Carl Zeiss Ltd., Cambridge, UK).

### 2.7. Quantification of Consumption and Release of Key Metabolites (CORE Analysis)

The quantification of pyruvate and lactate in the conditioned maturation medium, and glucose, pyruvate, and lactate in spent SOF analysis media, was performed using a microfluorometric assay as described previously [31]. Briefly, 1 μL of sample or blank medium was added to 9 μL of relevant assays’ mixture, and the concentration of the key metabolites was determined indirectly using the enzymatic-induced oxidation/reduction of NAD(P)H, which was measured fluorometrically. The pyruvate assay mixture comprised 0.1 mM NADH and 40 IU/mL lactate dehydrogenase in 4.6 mM EPPS buffer, pH 8.0, which was co-incubated with samples at 37 °C for 3 min. The final concentrations in each sample were determined against a six-point standard curve from 0–0.45 mM pyruvate for BMM media and 0–0.36 mM for SOF analysis media. The lactate mixture contained 40 IU/mL lactate dehydrogenase (LDH) in a glycine-hydrazine buffer, pH 9.4, which was co-incubated with samples at 37 °C for 30 min. A six-point standard curve of lactate was used to determine the lactate concentration in the spent media. A 0 to 2.5 mM and a 0 to 1.25 mM standard curve was run against BMM samples and SOF analysis samples, respectively. Finally, the glucose mixture included 0.4 mM dithiothreitol, 3.07 mM MgSO_4_, 0.42 mM ATP, 1.25 mM NADP^+^, 20 IU/mL hexokinase/glucose-6-phosphate dehydrogenase (HK/G6PDH) in EPPS buffer at pH 8.0. Samples were co-incubated at 37 °C for 10 min. The final concentration of glucose in the SOF analysis media was calculated based on a six-point, 0 to 0.5 mM, standard curve. After the final concentration was calculated, further normalisation was performed against the duration of incubation for COCs or blastocysts and the number of COCs within a droplet. Blastocysts were cultured singly. Data are expressed in pmol/hour/COC or pmol/hour/embryo, respectively.

### 2.8. Quantification of Amino Acid Turnover

Reverse-phase high-performance liquid chromatography was used as described previously [32] for the quantification of 18 amino acids in the spent SOF analysis media. Briefly, the detection of the amino acid is based on the different sizes and hydrophobicities of the amino acids. The amino acids underwent pre-column derivatisation with O-phthaldialdehyde (OPA), supplemented with 1 mg/mL 2-mercaptoethanol, which resulted in the formation of amino acid conjugates with OPA and generated a fluorescent signal, detectable at 450 nm.

Aspartic acid, glutamic acid, asparagine, serine, histidine, glutamine, glycine, threonine, arginine, alanine, tyrosine, tryptophan, methionine, valine, phenylalanine, isoleucine, leucine, and lysine were quantified with this technique calculated using the retention time (RT) and the area under the curve (AUC), respectively, and compared to standards. Certified amino acid standards were used (AAS18, Sigma Aldrich, Burlington, MA, USA) supplemented with asparagine, glutamine, tryptophan, and 2-Diethylomino-N-3-phenylmethoxypheny-l-acetamide (D-ABA), which were made in-house. D-ABA was used as an internal standard, as a non-metabolisable amino acid. The concentration of the amino acids in the final standard solution was 12.5 μΜ. The samples were diluted to 1:12.5 in ddH_2_O and analysed with a single injection per vial.

The concentration of each amino acid in each sample was further subtracted from the concentration of the same amino acid in the blank drop, and finally normalised against the recorded embryo incubation time, and the results were expressed as pmol/hour/embryo.

### 2.9. Determination of Mitochondrial Function Using Oxygen Consumption Rate

The mitochondrial function after acute TCS exposure of oocytes prior to maturation was determined using the measurement of oxygen consumption rate (OCR) and its components with extracellular flux analysis (EFA) as reported by [45]. Briefly, using a sensor-containing Seahorse fluxpak, OCR and its components were measured in a Seahorse-XFp analyser (Agilent, Santa Clara, CA, USA). For each assay, an overnight pre-incubation in a non-CO2 humidifier and an ‘on-the-day calibration’ of the sensor-containing Seahorse fluxpak were included. Upon completion of these steps, COCs were loaded into the cell plate and placed into the Seahorse-XFp.

For the purpose of this experiment, a 17-point measurement protocol was developed. Each point of measurement included a three-minute measurement and a one-minute wait period. During the three-minute measurement period, the sensor was lowered, creating an airtight microenvironment able to detect the depletion of dissolved oxygen. During the one-minute waiting step, the sensor was raised up, allowing recalibration. During the course of the 17-point protocol, 4 serial injections occurred: 1 TCS and 3 mitochondrial inhibitors (oligomycin, carbonyl cyanide 4-(trifluoromethoxy)phenylhydrazone (FCCP), and Rotenone/Antimycin A). During the first 3 points, no injection took place, TCS was injected after the completion of the third point, oligomycin after the eighth point, FCCP was added after the eleventh point, and a mixture of Rotenone/Antimycin A was injected after the fourteenth point. The final concentration of TCS in the cell wells was 0, 1, or 10 nM, depending on the group, 1 μM for oligomycin, 5 μM FCCP, and finally 2.5 μM of Rotenone/Antimycin A mixture as suggested by Muller et al. [45]. The results of the OCR assay, expressed as pmol/min/well, were normalised to oocyte number per well.

### 2.10. Statistical Analysis

All data were normalised against the number of the COCs or embryos and the hours of incubation, as described above in each section of the methods. Proportional data underwent Arc-sine transformation prior to any statistical test. The normal distribution of each non-proportional data set was checked using the D’Agostino and Pearson test or the Shapiro–Wilk test in the cases where the sample size was too small for the D’Agostino and Pearson test. The data sets that failed the normality test were log transformed, and parametric tests (one-way ANOVA or two-way ANOVA) were used to identify possible differences. In the cases where log transformation was not possible, non-parametric tests were used. One-way ANOVA followed by post-hoc Tukey test or Kruskal–Wallis test followed by Dunn’s test were used when more than two groups were compared. Two-way ANOVA followed by post-hoc Tukey test or multiple Mann–Whitney were used when each experimental group included more than one condition. All statistical tests were performed on GraphPad Prism (V9.0.1) using a significant difference *p* value < 0.05. All graphs are represented as mean ± SD.

## 3. Results

COCs were first exposed to two different concentrations of TCS during maturation and hormone release prior to the determination of metabolic activity. Release of E2 by bovine COCs in vitro was not influenced by the presence of TCS regardless of the administered dose (Figure 1a). P4 production was not influenced by the high dose of TCS (10 nM); however, COCs exposed to 1 nM TCS released significantly more progesterone than non-treated controls (2.85 vs. 4.78 pg/hour/oocyte *p* = 0.034; Figure 1b).

Next, the energy substrates, pyruvate production/depletion, and lactate production by COCs during IVM were recorded (Figure 2). The findings revealed that COCs cultured without TCS or with the vehicle solvent (DMSO) released pyruvate (82.84 ± 33.9 pmol/hour/COCs and 73.97 ± 56.1 pmol/hour/COCs, respectively) whilst pyruvate was depleted from culture media by COCs when exposed to TCS *p* < 0.0001 (Figure 2a). COCs exposed to a 10 nM dose consumed an average of 112.8 ± 35.8 pmol/hour/COCs of pyruvate, and those exposed to 1 nM exhibited a mean pyruvate consumption of 106.4 ± 27.1 pmol/hour/COCs. Notably, TCS had no effect on lactate release (Figure 2b).

In addition, the nuclear status of 352 COCs in total after IVM was assessed (Table 1). The values corresponded to three biological replicates. TCS exposure did not affect the proportion of COCs reaching MII stage of maturation (Figure 3).

Since lactate production was not influenced, and the nuclear status of COCs after IVM was not significantly impacted by the presence of TCS, it was possible that the increased depletion of pyruvate by oocytes exposed to TCS may reflect an increased demand for energy production to counterbalance the effects of the presence of TCS. To test this hypothesis, OCR and its components were measured on COCs during acute exposure to TCS prior to IVM.

The effect of TCS on bioenergetic profiles is presented in Figure 4. Basal OCR by COCs without TCS was first established before TCS was injected. The OCR response to TCS was compared to basal respiration (Figure 4c). Subsequently, oligomycin, as an ATP-synthase inhibitor, was injected into all groups, revealing the proportion of OCR which is coupled to ATP generation. This was followed by the addition of FCCP, which uncouples oxygen consumption from oxidative phosphorylation and thus indicates maximal OCR. Finally, Antimycin A and Rotenone (A/R) were added in combination. The OCR insensitive to A/R is related to non-mitochondrial activity (Figure 4a). The response to the TCS treatment and to the mitochondrial inhibitor, per experimental group, is illustrated in Figure 4b.

Upon addition of fresh culture medium to the control, there was a positive change (modest increase) in OCR, likely due to the replenishment of nutrients. However, this increase was ablated when 10 nM TCS was added, with a modest impairment of OCR (Figure 4c). By contrast, 1 nM TCS had no effect on total OCR (Figure 4c).

The origin of the OCR difference in response to the treatment among the groups is presented in Figure 4d,e. The components of OCR were calculated as a percentage of COCs’ basal respiration after the treatment’s injection. The proportion of OCR directly coupled to ATP synthesis was 58.7 ± 5.2% for the control group, whilst for 10nM-TCS and 1nM-TCS, it was 33.1 ± 2.5% and 33.3 ± 5.6%, respectively. In other words, oxygen consumption directly coupled to ATP synthesis was decreased by approximately 25% when the COCs were exposed to TCS, irrespective of the administered dose; a statistically significant difference compared to the control group (Figure 4c). Moreover, the proportion of the OCR linked to non-mitochondrial activity was significantly increased when the COCs were exposed to a 10nM-TCS dose (28.4 ± 3%) compared to the control group (10.5 ± 3.2%). Finally, a strong trend of increased proton leak was observed when the COCs were exposed to the 1 nM-TCS dose compared to the control group (*p* = 0.065).

TCS-treated COCs were next fertilised, and embryos cultured without further addition of TCS. Rates of cleavage and blastocyst formation were assessed on Day 2 and Day 7 to 8, respectively. The results showed that the cleavage rate of bovine embryos produced from TCS-exposed COCs was not significantly different from the control group (Figure 5a). However, fewer blastocysts were formed from COCs exposed to 1 nM TCS compared to the control group (12.22 vs. 29.11% *p* = 0.02; Figure 5b). In addition to the reduced blastocyst rate in embryos derived from COCs exposed to 1 nM TCS, the observations suggest that the day of blastocyst formation was slightly delayed compared to the control group (Figure 5c); however, this difference was not statistically significant. Future morphokinetic studies could possibly identify a more subtle difference in the timing of development.

Based on these findings, the metabolic profile of the blastocysts was also determined. To enable this, blastocysts were cultured in individual drops for 22 to 24 h. The key cellular substrates, glucose, pyruvate, and lactate, which are the main energy substrates during preimplantation embryo development, as well as 18 amino acids, were all measured in spent media using a microfluorometric assay and HPLC, respectively.

Glucose consumption did not differ significantly between blastocysts derived from TCS-exposed COCs compared to the control group (Figure 6a). Further comparison of glucose depletion between blastocysts that developed and those that did not develop further during the 24 h of individual culture showed, as was expected, that the developed blastocysts consumed a higher amount of glucose per hour compared to those that did not develop. The difference in glucose consumption was statistically significant among the developed and non-developed blastocysts derived from the 10 nM-TCS exposed COCs (Figure 6d).

However, the blastocysts derived from COCs treated with 10 nM-TCS or 1 nM-TCS consumed almost double (1.96×) the amount of pyruvate compared to the blastocysts derived from the untreated COCs (−19.86 vs. −10.12 pmol/h/embryo *p* = 0.011 and −21.41 vs. −10.12 pmol/h/embryo, respectively) (Figure 6b). Further analysis of these results shows that pyruvate production was significantly increased in the blastocyst group, which remained in the same developmental stage during individual culture (not developed) (Figure 6e).

Finally, lactate production by blastocysts produced from the 10 nM-TCS exposed COCs was significantly reduced compared to controls and to the low dose of TCS group (15.20 vs. 31.82 pmol/h/embryo and 15.20 vs. 30.38 pmol/h/embryo, respectively) (Figure 6c). However, the blastocysts with reduced lactate production had not progressed in their development during the preceding 24 h of individual culture (Figure 6f).

Additionally, using the spent media after 24 h of individual blastocyst culture, amino acid turnover was determined by HPLC (Table 2). The data revealed that, out of 18 amino acids measured, asparagine, glycine, and leucine concentrations in the spent medium of blastocysts from TCS-exposed COCs were significantly different from controls (Figure 7). Blastocysts derived from TCS-exposed COCs depleted asparagine and released glycine, whilst the control groups released asparagine and consumed glycine. Interestingly, the low dose of TCS (1 nM) seemed to have a stronger effect on the metabolic state of those two amino acids compared to the higher dose (10 nM). Finally, leucine consumption was significantly decreased by blastocysts derived from COCs treated with the high dose of TCS compared to 1 nM-TCS, with a strong trend towards depletion compared to the control group.

## 4. Discussion

In this study, we have used a bovine model to demonstrate that exposing oocytes to TCS leads to altered steroidogenesis and disrupted metabolic and mitochondrial function. In addition, embryos made from oocytes exposed to TCS exhibit significantly altered usage of metabolic substrates, indicative of further metabolic adaptation. The concentrations of TCS (1 and 10 nM) administered are representative of TCS levels identified in human bodily fluids [6,8,10,11,13,16], which, in the majority of cases, occur in the nanomolar range.

Oocyte quality and the environment in which development occurs are major determinants of ongoing embryo viability. The subsequent preimplantation stages of development are of critical importance since impairment during this period of development can have detrimental effects on pregnancy outcomes and, of greater concern, on the lifelong health of the offspring, captured cogently by the Developmental Origins of Health and Disease hypothesis [39]. According to this hypothesis, exposure of gametes and preimplantation embryos to chemicals and other environmental agents can alter the developmental fate of an organism.

Due to industrialization and modern lifestyle changes, humans and other organisms are exposed to diverse classes of chemicals either indirectly due to their accumulation in the environment [47] and food [48,49] or directly through the use of consumer products [50]. A typical example of this is TCS, a chemical that has been used in many personal hygiene products as an antimicrobial agent [51] and may accumulate in the environment due to its excessive use [51,52]. Thus, as mentioned earlier, TCS has been identified in human biological fluids such as urine, serum and breast milk, and ovarian follicular fluid [16].

In addition to the results presented herein, a spectrum of in vitro and in vivo studies using TCS exposures have shown detrimental effects on different physiological endpoints [51,53], including the reproductive system [54,55]. Specifically, such studies have revealed that TCS acts as an Endocrine-Disrupting Chemical (EDC) and can interfere with E2, P4, and thyroxine production due to its similarity with E2 [56]. It is within this context that TCS has been reported to affect implantation and placental function in a range of animal models, as reviewed expertly in [19]. Despite these findings, the impact of TCS exposure on mammalian oocyte maturation and the early stages of embryo development remains relatively unknown. Two of the very few papers in this area [20,21] examined TCS toxicity in porcine oocytes and reported results, which aligned with our own findings in suggesting that TCS exposure during in vitro maturation impairs mitochondrial function in porcine oocytes. It has also been reported that preimplantation embryo development in pigs [57] and mice [26] is influenced by the presence of TCS. Despite growing evidence that TCS has a negative impact on early reproductive events, due to the use of a wide range of administered concentrations and periods of exposure that have been used, the precise mechanism of action of TCS remains unclear.

The results of the present study revealed that bovine COCs treated with 1 nM TCS released significantly more progesterone than controls. Similar observations have been reported previously for human [58] and rat [59] granulosa cells in a range of different TCS concentrations (1 nM–10 μM). However, when rats were supplemented with TCS in food at 50 mg/kg/day for 28 consecutive days, progesterone levels in serum fell significantly [60]. A decrease in P4 levels has been reported in mice in vivo exposed to 10 and 100 mg/kg/day [61].

Moreover, exposure to TCS (1 nM–10 μM) has been reported to increase production of E2 in human and rat granulosa cells [58,59]. This contrasts with our observations; E2 production during in vitro maturation of bovine COCs remained unaffected by TCS. However, our results align with findings from whole animal dose studies in rats [60] and mice [61], which failed to see any differences in E2 levels in serum in response to TCS.

The apparent differences observed in our study, providing evidence for an influence of TCS on P4 and E2 levels, may be explained by the duration of the exposure or the animal model that has been used. In addition, these differences may arise from different administered doses, indicating a nonmonotonic action of TCS. The nonmonotonic action of TCS can further be supported by our results, as the 1 nM dose affected P4 production; however, this remained unchanged when the COCs were exposed to 10 nM TCS. The nonmonotonic action of EDCs has previously been reported [62], notably for BPA [63]. TCS has a similar chemical structure to BPA, both of which exhibit estrogenic action and structural similarity to E2. Importantly, our findings align with the conclusion of studies suggesting the endocrine-disrupting properties of TCS. However, further studies on the non-monotonic action of TCS need to be done.

Beyond steroidogenesis, exposure to TCS significantly affected the metabolic status of COCs in culture. COCs exposed to 1 or 10 nM TCS after IVM depleted pyruvate from the culture medium, whereas controls released pyruvate. Pyruvate is an essential energy substrate during preimplantation embryo development [38]. Furthermore, pyruvate is required for the completion of meiotic maturation in mouse [64,65]. In a study conducted in mouse COCs using maturation media supplemented with FSH during IVM, pyruvate was produced by COCs that had reached the MII stage but was consumed by nuclear immature COCs [66]. Pyruvate is also produced as the product of glycolysis in the cytoplasm, from where it is either converted to lactate or enters the mitochondria and is converted to acetyl-CoA, which enters the TCA cycle. Based on our observations of altered pyruvate handling by COCs treated with TCS, we proposed that these COCs were either entering meiotic arrest or displaying mitochondrial dysfunction. To exclude meiotic arrest as the cause for disrupted pyruvate metabolism, the nuclear status of COCs exposed to TCS was determined. The results confirmed that maturation status was not influenced by TCS treatment. We next examined mitochondrial function by determining oxygen consumption rate (OCR). Immediately after aspiration, COCs were treated with 0, 1, or 10 nM of TCS, and the OCR was determined. In addition, mitochondrial function was examined using the mitochondrial inhibitors oligomycin and Antimycin/Rotenone mix, which inhibit Complex V, and Complex III/Complex I, respectively, as well as FCCP, which acts as a mitochondrial uncoupler [45]. The results revealed that the total OCR was significantly reduced in COCs exposed to a 10 nM dose of TCS. Further analysis of this result showed that the ATP-coupled OCR, when expressed as a percentage of the total OCR, was significantly reduced in TCS-treated COCs compared to the untreated group. Mitochondrial dysfunction due to decreased ATP levels after TCS exposure has also been reported recently in porcine oocytes [21]. In addition, the rate of non-mitochondrial OCR was significantly increased after 10 nM TCS exposure, and a strong increase was observed in the proportion of OCR devoted to proton leak after COCs exposure to 1 nM TCS. These results confirm the mitochondrial dysfunction after TCS exposure of bovine COCs during IVM. TCS mitochondrial toxicity has previously been confirmed in different mammalian cells, including porcine sperm (using exposure levels of 1–10 μg/mL; [67]), human granulosa cells [58], and porcine oocytes and embryos [20,57] after direct exposure in the micromolar range (from 1 to 100 μΜ).

To investigate whether onward oocyte developmental competence is impaired after TCS exposure, the impact on the early stages of embryo development of TCS-exposed COCs was investigated. COCs treated with TCS were fertilised and cultured for seven days without further exposure to TCS. Our findings revealed that blastocyst rate was significantly reduced in the 1 nM-TCS group compared to the control group (12.22 vs. 29.11% *p* = 0.0205). However, a dose of 10 nM-TCS did not influence the developmental rates of the embryos. Reduced blastocyst rates due to TCS exposure during preimplantation stages of development have recently been confirmed in a study using the porcine as a model [57]. In that study, mitochondrial dysfunction was associated with a low blastocyst rate, which was partially confirmed by our results, where we observed that both doses of TCS impaired mitochondrial function, but the 1 nM dose of TCS only had an impact on blastocyst rate.

Finally, the metabolic status of the blastocysts generated in each group was assessed by transferring them into an individual drop culture system for 24 h. The blastocysts derived from TCS-treated COCs consumed significantly more pyruvate, irrespective of the administered dose, compared to the control group. Meanwhile, the lactate production of blastocysts derived from 10 nM-TCS COCs was significantly reduced. Further analysis showed that pyruvate depletion was higher compared to the control group, regardless of the development of blastocyst during the 24 h of individual culture. Finally, the amino acid turnover of the blastocysts generated was influenced in that 3 out of 18 amino acids were significantly different among the groups.

Overall, our results show that the metabolic activity of blastocysts derived from TCS treatment was in a “hyperactivated mode” in that they consumed more pyruvate and increased the turnover of the amino acids. One interpretation of these data is that embryos derived from COCs treated with TCS increase their metabolic function in order to counterbalance the effects of TCS. Embryos with higher metabolic activity in terms of pyruvate consumption [31], amino acid turnover [32,33], and oxygen consumption [36,68] have previously been associated with lower developmental potential [69]

Human females with subfertility may use Assisted Reproductive Techniques (ART); however, the aetiology of about 15% of the infertility cases remains unclear [70]. Subfertility or infertility of human females at reproductive age has been correlated with a wide range of causes, including exposure to natural or synthetic chemicals [71]. The bovine has previously been suggested as a suitable model for human IVF [43] and to evaluate the effect of chemicals on toxicological studies in reproduction [44] due to the biochemical and developmental similarities during the final stages of oocyte maturation, fertilisation, and early stages of embryo development. As mentioned earlier, TCS, due to its excessive use, is accumulated in the environment, resulting in people receiving direct or indirect exposure. For this reason, the present study was conducted to test TCS effects on oocyte and embryo metabolism using the bovine as a model for humans.

## 5. Conclusions

Overall, this study has found that TCS exposure during IVM impairs the metabolic status of COCs by affecting the progesterone production and mitochondrial function, resulting in reduced developmental competence. The COCs’ altered metabolic function not only reduced the developmental rates but also interfered with the metabolic activity of the embryos produced, resulting in reduced developmental potential. These findings are of special concern, taking into consideration that the levels of TCS chosen for the present study reflect levels common in everyday exposures for humans. Crucially, the implications of these findings in the longer term are not known but may relate to adverse effects on pregnancy outcomes or on health in later stages of development. Future studies could examine the action of TCS and related compounds on embryo morphokinetics and metabolism during the early stages of embryo development and during the onset of implantation and the impact of this exposure on later stages of life.

## Figures and Tables

**Figure 1 ijerph-22-01031-f001:**
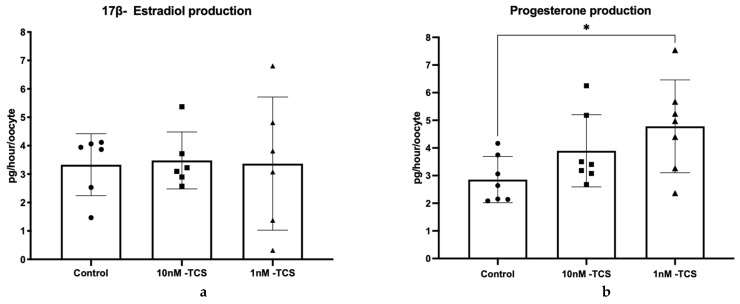
Release of E2 (**a**) and P4 (**b**) from bovine COCs exposed to two different concentrations of TCS at 10 or 1 nM during in vitro maturation (*n* = 6; each dot represents an individual replicate). Differences were assessed using one-way ANOVA followed by the Tukey test. * *p* < 0.05. Data are presented as mean ± SD.

**Figure 2 ijerph-22-01031-f002:**
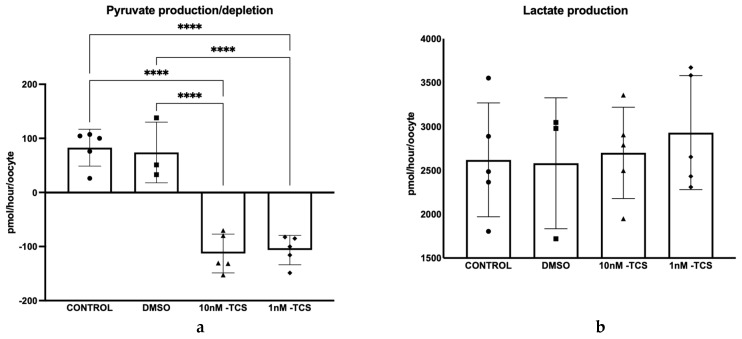
Depletion/release of pyruvate (**a**) and release of lactate (**b**) from bovine COCs exposed to two different concentrations of TCS at 10 or 1 nM during in vitro maturation (*n* = 5 individual replicates with 25–35 oocytes per replicate). Differences were tested for significance using one-way ANOVA followed by the Tukey test. **** *p* < 0.0001. Data are presented as mean ± SD.

**Figure 3 ijerph-22-01031-f003:**
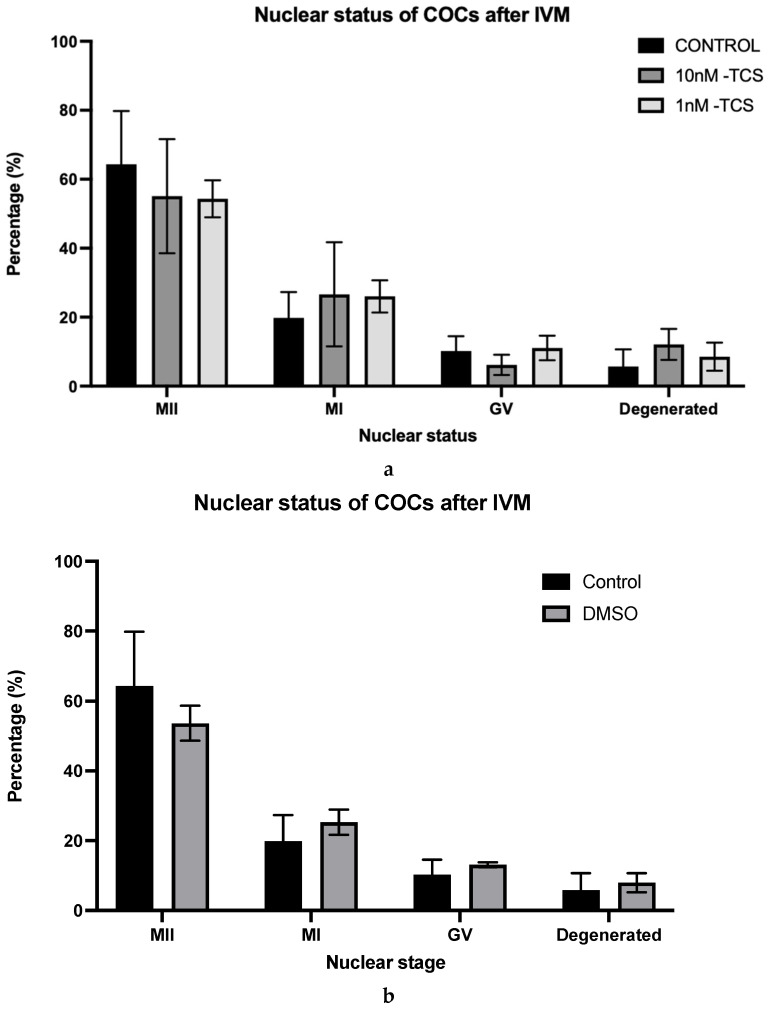
Nuclear status of bovine COCs exposed to TCS at 10 or 1 nM during IVM (**a**) or DMSO Vehicle control (**b**). Two-way ANOVA followed by a Tukey test was performed. (*n* = 3; 352 oocytes in total).

**Figure 4 ijerph-22-01031-f004:**
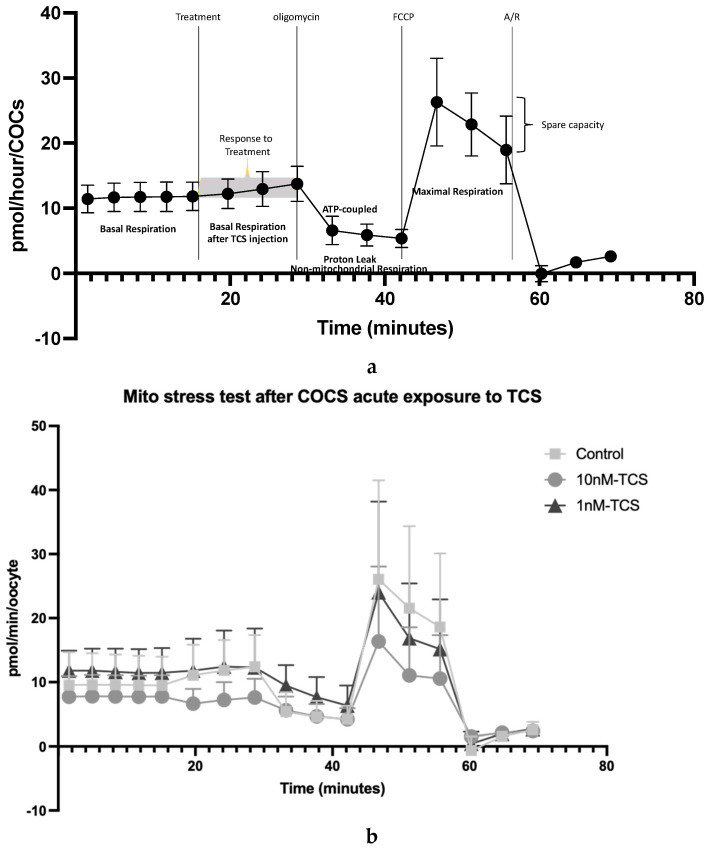
(**a**,**b**) OCR in COCs in response to TCS, oligomycin, FCCP, and A/R (**a**) as indicative examples of mitochondrial parameters and (**b**) as comparison among the different treatments, (**c**) OCR as response to the TCS treatment, and (**d**,**e**) mitochondrial parameters as a percentage of basal OCR after TCS injection. Differences were detected using (**c**) one-way ANOVA followed by a Tukey test. (**e**) Two-way ANOVA followed by a Tukey test. ns = not significant * *p* < 0.05; ** *p* < 0.01; (OCR, *n* = 3).

**Figure 5 ijerph-22-01031-f005:**
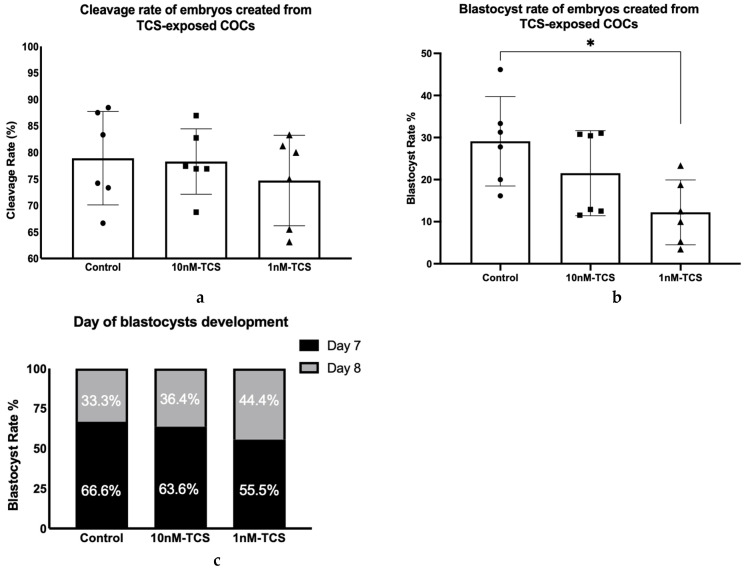
(**a**) Cleavage rate and (**b**) blastocyst rate of embryos produced by TCS-exposed COCs during IVM. (**c**) The day of blastocyst formation is expressed as the percentage of the total number of blastocysts per group. Differences were tested using one-way ANOVA and a post hoc Tukey test. (*n* = 6 biological replicates); * *p* < 0.05.

**Figure 6 ijerph-22-01031-f006:**
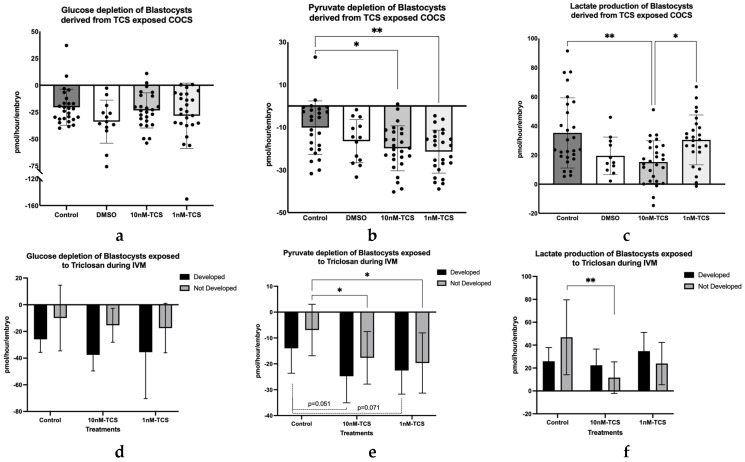
(**a**) Glucose and (**b**) pyruvate depletion and (**c**) lactate production of the total number of blastocysts produced per treatment group. Each dot in graphs (**a**–**c**) represents the energy substate’s production or depletion of an embryo. Comparison of (**d**) glucose (**e**) pyruvate depletion and (**f**) lactate production by blastocysts whose development progressed (developed) or did not progress (not developed) during individual embryo culture. The distribution of each data set was tested; in those cases in which the distribution was normal, a one-way ANOVA followed by a Tukey test was performed, (**b**) while the difference between developed vs. not developed was tested using two-way ANOVA (**e**). Kruskal–Wallis test followed by Dunn’s test was used when the data set failed in the normal distribution test (**a**,**c**). In this case, the differences among developed vs. not developed blastocysts were tested using the Kruskal–Wallis test followed by Dunn’s and Mann–Whitney (**d**,**f**). * *p* < 0.05, ** *p* < 0.01. Data presented in pmol/h/embryo ± SD.

**Figure 7 ijerph-22-01031-f007:**
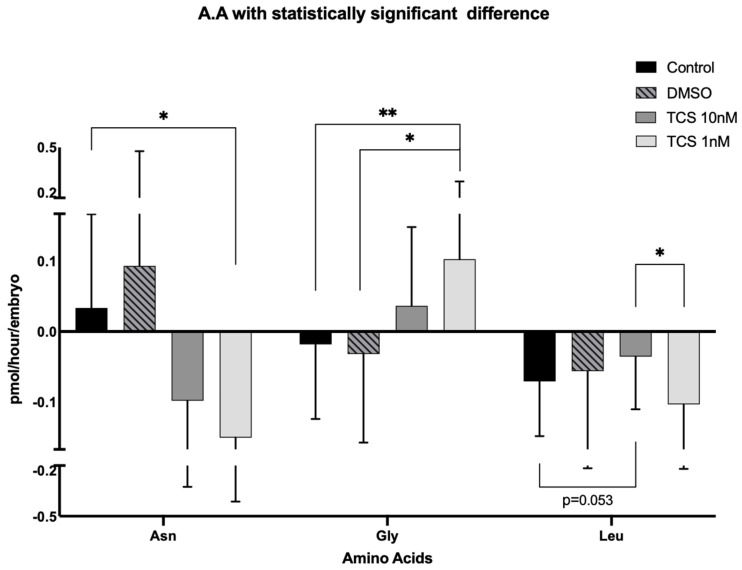
Amino acid turnover of blastocysts derived from TCS-exposed COCs. The graph depicts the 3 out of 18 amino acids, asparagine (Asn), glycine (Gly), and leucine (Leu) with significant differences after statistical analysis using one-way ANOVA, followed by post hoc analysis with Tukey or Kruskal–Wallis test followed by Dunn’s test depending on whether the amino acid followed a normal distribution or not. * *p* < 0.05, ** *p* < 0.01. The results are expressed as pmol/h/embryo ± SD.

**Table 1 ijerph-22-01031-t001:** Number of COCs assessed for the nuclear status per group of treatment.

Total Number of COCs Assessed for Nuclear Status	Treatment Group			
	Control	DMSO	TCS (10 nM)	TCS (1 nM)
352	85	99	87	81

**Table 2 ijerph-22-01031-t002:** Turnover of each individual amino acid per treatment group. Amino acids in bold with an asterisk * are those with statistically significant difference (*p* < 0.05). The results are expressed on pmol/h/embryo ± SD The amino acids * are illustrated in Figure 7.

Amino Acids	Treatment							
	Control		DMSO		TCS (10 nM)		TCS (1 nM)	
	Mean	SD	Mean	SD	Mean	SD	Mean	SD
Asp	0.068	±0.147	0.113	±0.237	0.043	±0.273	0.025	±0.178
Glu	−1.031	±0.771	−1.083	±0.727	−1.209	±0.943	−1.32	±0.745
**Asn ***	**0.035**	**±0.139**	**0.093**	**±0.382**	**−0.113**	**±0.201**	**−0.15**	**±0.253**
Ser	−0.086	±0.079	−0.098	±0.111	−0.074	±0.069	−0.091	±0.220
His	−0.059	±0.160	0.073	±0.287	−0.042	±0.104	−0.099	±0.152
Gln	0.179	±0.484	0.656	±1.018	0.274	±0.646	0.254	±1.228
**Gly ***	**−0.013**	**±0.109**	**−0.032**	**±0.125**	**0.036**	**±0.111**	**0.102**	**±0.173**
Thr	−0.016	±0.088	−0.02	±0.075	−0.033	±0.105	0.038	±0.234
Arg	−0.729	±0.556	−0.742	±0.697	−0.669	±0.449	−0.455	±1.117
Ala	0.865	±0.492	0.765	±0.576	0.748	±0.650	0.981	±0.713
Tyr	−0.016	±0.053	−0.034	±0.056	−0.017	±0.053	−0.009	±0.071
Trp	−0.032	±0.049	−0.026	±0.057	−0.034	±0.062	−0.061	±0.075
Met	−0.031	±0.027	−0.03	±0.043	−0.018	±0.024	−0.029	±0.025
Val	−0.037	±0.055	−0.039	±0.072	−0.029	±0.056	−0.032	±0.059
Phe	−0.013	±0.025	−0.012	±0.042	−0.006	±0.026	−0.011	±0.029
Iso	−0.03	±0.040	−0.028	±0.06	−0.028	±0.040	−0.034	±0.047
**Leu ***	**−0.086**	**±0.071**	**−0.056**	**±0.128**	**−0.027**	**±0.078**	**−0.103**	**±0.086**
Lys	0.215	±0.274	0.383	±0.462	0.262	±0.300	0.261	±0.276

## Data Availability

Original data is available on request of the corresponding author.

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
