# Peer review of "Oocyte Exposure to Low Levels of Triclosan Has a Significant Impact on Subsequent Embryo Physiology"

_ijerph, 2025, doi:10.3390/ijerph22071031_

Round 1

Reviewer 1 Report

Comments and Suggestions for Authors

The article is about a very important topic, since we are exposed to this compound on a daily basis and its consequences for human health are still little explored and clear. I therefore consider the topic to be very original. 
However, it does not mention the introduction of a crucial aspect of this compound, which is its endocrine disrupting action (Int J Mol Sci
. 2022 Sep 28;23(19):11427. doi: 10.3390/ijms231911427). Therefore, the introduction should address this very important topic, without which the aim of this work would not be as relevant. 
With regard to the discussion and despite having a sentence on this effect, it should be explored. Remember that this action is much more important in the most susceptible population, namely pregnant women and their foetuses/embryos, where the consequences are much more visible.

Author Response

Comment 1: Inclusion of TCS's role as EDC.

response: We are grateful to the reviewer for highlighting the relevant article and have included this with a slight expansion in the introduction (lines 56-58) and discussion (lines 524-526). However, as the article is already quite lengthy, we are reluctant to expand too much as this risks making the paper too long for readers.

Reviewer 2 Report

Comments and Suggestions for Authors

Author Response

Comment: Numerous minor and typographical suggestions in the attached document

Response: We are extremely grateful to this reviewer for their dedicated and detailed review of our paper, highlighting a number of minor typographical errors.  Each of the points raised have been addressed and can be seen in the track-changed version of the modified manuscript.

Reviewer 3 Report

Comments and Suggestions for Authors

Papachristofi et al assess the impact of TCS on mammalian cumulus-oocyte-complexes. Scientifically I think the work is sound and I dont really have any major concerns about it. My minor comments would be that the first two paragraphs of the introduction started to just feel like a long list of references where people had detected TCs in different cohorts of humans. I get the point of demonstrating it's real world relevance, but it felt like the point was over laboured in this instance. 

I also didnt understand why Figure 2 was broken into two separate graphs, especially as the control I believe is the same for each graph?

It also felt slightly odd to have the orientation of the higher dose then the lower dose in all of the graphs. It just made interpreting the graphs a bit more challenging and I didnt really get the point of it.

Minor things:

typos: l86 (?) l109 (underline), l119 (i the), l120 (wnashed), l166 (-80oC), l231 (CO2)

Also feels like there are a few instances of double spaces as well. 

Author Response

Comment 1: Papachristofi et al assess the impact of TCS on mammalian cumulus-oocyte-complexes. Scientifically I think the work is sound and I dont really have any major concerns about it. My minor comments would be that the first two paragraphs of the introduction started to just feel like a long list of references where people had detected TCs in different cohorts of humans. I get the point of demonstrating it's real world relevance, but it felt like the point was over laboured in this instance. 

Response: Thank you for considering our manuscript; indeed, the first part of the paper is illustrating the scale of evidence describing the exposure of TCS.  Whilst we acknowledge that this may appear overly long, we do feel it important to include to give readers less familiar with the topic a good degree of background to understand the context of the study.

Comment 2: I also didnt understand why Figure 2 was broken into two separate graphs, especially as the control I believe is the same for each graph?

Response 2: this was presented for simplicity, but concede that the DMSO data could easily be presented as a supplementary figure, as this is a control, and not really relevant to the study.  I will leave it to the editor to decide whether this can go as a supplementary figure.

Comment 3: It also felt slightly odd to have the orientation of the higher dose then the lower dose in all of the graphs. It just made interpreting the graphs a bit more challenging and I didnt really get the point of it.

Response 3: Thank you for the comment.  We respect the perspective of the reviewer, however this was not raised by any of the other reviewers and we feel that the order of presentation helps to give the reader the impression of reducing concentrations used in our study.

Comment 4: Typos

Response 4: We would like to thank this reviewer for raising some of these issues.  We have carefully proof read the manuscript and have addressed all typos.